# Efficacy and Immunogenicity of a Recombinant Vesicular Stomatitis Virus-Vectored Marburg Vaccine in Cynomolgus Macaques

**DOI:** 10.3390/v16081181

**Published:** 2024-07-24

**Authors:** Vidyleison N. Camargos, Shannan L. Rossi, Terry L. Juelich, Jennifer K. Smith, Nikos Vasilakis, Alexander N. Freiberg, Rick Nichols, Joan Fusco

**Affiliations:** 1Department of Pathology, University of Texas Medical Branch, Galveston, TX 77555, USA; vinevesc@utmb.edu (V.N.C.); tljuelic@utmb.edu (T.L.J.); jeksmith@utmb.edu (J.K.S.); 2Departments of Pathology, Microbiology and Immunology, Institute of Human Infection and Immunity, University of Texas Medical Branch, Galveston, TX 77555, USA; anfreibe@utmb.edu; 3Departments of Pathology, Center for Vector-Borne and Zoonotic Diseases, Institute of Human Infection and Immunity, University of Texas Medical Branch, Galveston, TX 77555, USA; nivasila@utmb.edu; 4Public Health Vaccines, LLC, Cambridge, MA 02142, USA; rnichols@phvaccines.com

**Keywords:** Marburg virus, filovirus, VSV, vesicular stomatitis virus, vaccine, pseudotyped vector, cynomolgus macaque, efficacy

## Abstract

Filoviruses, like the Marburg (MARV) and Ebola (EBOV) viruses, have caused outbreaks associated with significant hemorrhagic morbidity and high fatality rates. Vaccines offer one of the best countermeasures for fatal infection, but to date only the EBOV vaccine has received FDA licensure. Given the limited cross protection between the EBOV vaccine and Marburg hemorrhagic fever (MHF), we analyzed the protective efficacy of a similar vaccine, rVSV-MARV, in the lethal cynomolgus macaque model. NHPs vaccinated with a single dose (as little as 1.6 × 10^7^ pfu) of rVSV-MARV seroconverted to MARV G-protein prior to challenge on day 42. Vaccinemia was measured in all vaccinated primates, self-resolved by day 14 post vaccination. Importantly, all vaccinated NHPs survived lethal MARV challenge, and showed no significant alterations in key markers of morbid disease, including clinical signs, and certain hematological and clinical chemistry parameters. Further, apart from one primate (from which tissues were not collected and no causal link was established), no pathology associated with Marburg disease was observed in vaccinated animals. Taken together, rVSV-MARV is a safe and efficacious vaccine against MHF in cynomolgus macaques.

## 1. Introduction

Marburg virus (MARV) is a negative-stranded RNA virus in the family Filoviridae. This virus is a close relative to the more well-known Ebola virus (EBOV). Marburg virus (MARV) infection can result in severe hemorrhagic fever (MHF) in both people and non-human primates (NHPs), with fatality rates as high as 88% [1]. Although no large outbreaks have been recorded since 2004–2005, sporadic cases have been observed in other regions across the world [2,3]. Importantly, it is estimated that 105 million people across 27 countries live in areas suitable for MARV transmission [4]. Outbreak control strategies, as highlighted by the World Health Organization, include surveillance, early case identification and isolation, contact tracing, practicing appropriate burial procedures and local awareness campaigns [5,6]. Therefore, targeted filovirus countermeasures, including vaccines, antivirals and therapeutics, are critically needed for the management of MARV and MHF.

The development of vaccines against filoviruses has mixed success, especially for MHF. The main antigenic target for filoviruses is the viral glycoprotein, GP. Therefore, a variety of approaches to express GP antigen have been developed, including using viral vectors (recombinant vesicular stomatitis virus [rVSV] and adenovirus [rAdV]), virus-like particles (VLPs), DNA plasmids, and lipid-particle encoded RNA. Of all approaches, the VSV-vectored platform is one of the most attractive, as it combines an impressive safety profile, strong antigen presentation, replicates to high titers in vitro, which is advantageous compared to other viral-vectored vaccines, induces strong innate and adaptive immunity, and has been effective as a strategy for a variety of viral diseases (reviewed in [7]). In this system, the VSV fusion glycoprotein gene is deleted (rVSVΔG) and replaced with a gene of interest; for filovirus vaccines, the gene of interest is the GP [8]. This approach led to the first FDA-approved vaccine for Ebola virus disease (EVD), ERVEBO, in 2019 [9]. However, cross-reactivity is limited, making ERVEBO only effective against the EBOV Zaire virus and not other Ebola virus species or other filoviruses. Therefore, the need to create these vaccines is urgent.

NHP species, including rhesus and cynomolgus macaques (*Macaca fascicularis*), are considered the gold standard for vaccine efficacy studies, as they recapitulate hallmarks of human filovirus disease. Specifically, the Filovirus Animal Non-Clinical Group has defined the parameters and challenge strains for vaccine efficacy testing (summarized for EBOV and MARV [10,11]). The cynomolgus macaque provides a consistent and well-characterized model for MARV Angola infection, including fever, complete blood count (CBC) and serum chemistry, cytokines, viremia, and viral loads in key tissues and pathology [12,13,14].

Studies evaluating rVSV-MARV vaccine candidates against 2 MARV strains, Musoke and Angola, have reported high survival rates in vaccinated animals from lethal challenge models [8,15,16,17,18,19,20,21,22,23,24,25]. Among these studies, 100% survival was observed when animals were challenged as early as 7 days, or as late as 14 months post vaccination. Another study showed rVSV-MARV candidates are therapeutically active; high survival rates were observed when animals were treated 20–30 min post lethal MARV infection.

This study describes the proof-of-concept for a rVSVΔG-MARV-GP vaccine (referred to as rVSV-MARV) to provide complete protection against the MARV Angola infection in the cynomolgus macaque model. This vaccine was previously shown to be immunogenic and confer protection against MARV in the guinea pig model [26]. Here, two studies were staggered to test the efficacy of different vaccine lots and doses (Experiments 1 and 2), both yielding similar results. Briefly, all rVSV-MARV-vaccinated primates seroconverted to MARV-GP following a single dose of the vaccine, and were fully protected from lethal MARV challenge without major alterations to the biochemical markers associated with MHF. These results indicate rVSV-MARV has the potential to be an effective prophylactic vaccine against MHF and, following successful IND-enabling safety studies, it is now the first rVSV-MARV vaccine to be tested in humans (Phase 1-studies. NCT06265012).

## 2. Materials and Methods

### 2.1. Cynomolgus Macaques and Ethics Statement

Two groups of 8 *Macaca fascicularis* NHP, Mauritius origin, ranging between 2.93–4.8 years of age and 3–6.3 kg, were purchased from Worldwide Primates, Inc. (Miami, FL). NHP health records were reviewed by University of Texas Medical Branch (UTMB) veterinarians and confirmed negative for various infections/diseases including TB, herpes B virus, simian immunodeficiency virus (SIV), simian T lymphotrophic virus-1 (STLV-1), simian retrovirus (SRV) and *Trypanosoma cruzi*. Blood samples collected prior to shipment were confirmed negative for Marburg virus antibodies. NHPs were housed in the Animal Biosafety Level 2 (ABSL2) during their pre-study quarantine and vaccination phase. Between days 34 and 42, NHPs were transported to the ABSL4 for MARV challenge.

All procedures with cynomolgus macaques were performed under an approved Institutional Animal Care and Use Committee (IACUC) protocol. UTMB is an AALAC-accredited institution which strictly adheres to the Animal Welfare Act, and any federal and state guidelines for animal research. All NHPs were given health checks by a veterinarian prior to and upon arrival at UTMB. Clinical observations were made at least twice daily for all NHPs. A score chart detailing alterations in respiration, appetite, fecal output, activity, appearance, and hemorrhage was used to determine if additional observations or humane euthanasia was required. The euthanasia criteria, regardless of cumulative score, include dyspnea/agonal breathing, inability to freely move around the cage (failure to rise from laying down), and severe hemorrhage. All efforts to minimize pain and distress, that would not interfere with the course of vaccination or disease, were taken.

At the end of the study or when NHPs reached a humane endpoint (based upon the score sheet), NHPs were deeply anesthetized with ketamine followed by a lethal dose of chemical euthanasia solution injected by a veterinarian. Terminal blood samples were collected. A full necropsy of the thoracic and abdominal cavity was performed as soon as possible after death.

Prior to manipulations, including vaccination, challenge, blood collection and weight measurements, NHPs were anesthetized intramuscularly with ketamine. Weights were taken at most anesthetic events. Blood was collected by venipuncture, with a preference from the femoral vein, into vacutainers appropriate for the assay; serum separator tubes (Vacuette, GmbH, Monroe, NC, USA) were used for viremia, antibody quantification, and serum chemistry, and EDTA tubes (Vacuette and BD, Franklin Lakes, NJ, USA) were used for plasma collection for vaccinemia and whole blood CBC analysis.

### 2.2. Virus and Cells

Vero cells (E6) were purchased from ATCC American Type Culture Collection, Manassas, VA, USA, CRL-1586) and used for plaque and neutralization antibody assays. Briefly, cells were maintained in cell culture media (Dulbecco’s Modified Eagles Medium supplemented with 10% heat-inactivated fetal bovine serum [FBS] with 1% penicillin-streptomycin) at 37 °C in 5% CO_2_.

The same MARV Angola stock (Marburg virus/H.sapiens-tc/AGO/2005/Angola-200501379, NR-48866) was obtained from BEI Resources and used as the challenge for Experiments 1 and 2. Aliquots of unused challenge stocks were used for MARV PRNTs and the standard curve for RT-PCR assays.

### 2.3. rVSV-MARV Vaccine and Vaccination

rVSV-MARV (PHV01), based upon the Angola strain, was previously produced and efficacy tested in guinea pigs [26]. rVSV-MARV was manufactured by and received from National Resilience Government Services, (formerly, Ology Bioservices, Inc. (Alachua, FL, USA)) in two different lots; Experiment 1 tested Lot#PTR4.1-15Oct2020 and Experiment 2 tested Lot#155941, both stored at −80 °C until use. At vaccination, vials were thawed on ice, mixed by inversion 5 times and loaded into 1 mL tuberculin syringes. All vaccines were kept on ice and used within 2 h of thawing.

On day 0, NHPs were anesthetized with ketamine and the fur above the quadriceps was shaved to allow for visualization at the injection site. The injection site was wiped down with 70% ethanol wipes prior to vaccination. In Experiment 1, both legs were injected with 1.0 mL of vaccine (total 2 mL, dose = 6.68 × 10^7^ pfu). In Experiment 2, the right leg was injected with 1.0 mL of vaccine (total 1 mL, dose = 1.6 × 10^7^ pfu). In both Experiments, control NHPs were injected with an equivalent volume of sterile 0.9% saline solution (Hospira, Lake Forrest, IL, USA).

### 2.4. MARV Challenge and Observations

On day 42 post vaccination, NHPs were anesthetized with ketamine and injected with 1.0 mL of MARV Angola stock diluted in Hanks buffered saline solution (HBSS), supplemented with 2% FBS to a concentration of 1 × 10^4^ pfu/mL in the right deltoid muscle. While the IM-delivered LD_50_ in cynomolgus macaques is not clearly quantified, it is estimated to be around 1 pfu [27], so the challenge dose delivered was ~10,000 × LD_50_. The diluted virus was maintained on ice during the dilution and injections, and was titrated by plaque assay after all injections to confirm titer. The backtiter of Experiments 1 and 2 were confirmed to be 1.29 × 10^4^ pfu and 1.48 × 10^4^ pfu, respectively.

Cynomolgus macaques were observed at least twice daily post challenge, and more frequently if a disease severity score was triggered on the filovirus clinical score sheet. All personnel taking observations were trained for signs of filovirus disease.

### 2.5. Virus Quantification

#### 2.5.1. Plaque Assay

Vero cells were seeded into monolayers in 12-well plates the day prior, then 10-fold serial dilutions of virus were diluted in media and inoculated onto cells for an hour. An overlay consisting of Eagle’s minimal essential media and 0.8% methylcellulose, 2% FBS, and 1% penicillin/streptomycin was added and plates incubated 37 °C under 5% CO_2_ for 15 days. Plaques were visualized on fixed monolayers by crystal violet staining.

#### 2.5.2. RT-qPCR for Quantification of MARV Viral RNA

MARV genomic equivalents were quantified by RT-qPCR. Viral RNA from experimental serum samples and known MARV stocks were extracted and inactivated in TRIzol LS according to manufacturer’s instructions (ThermoFisher Scientific, Waltham, MA, USA). The methods were previously described [28,29]. Briefly, MARV viral RNA was quantified using the StepOnePlus Real-time PCR System (Applied Biosystems) in One-Step Probe RT-PCR kits (Qiagen, Germantown, MD, USA) with the following cycle conditions: 45 °C for 10 min, 95 °C for 5 min, and 40 cycles of 95 °C for 5 s and 59 °C for 30 s. Primer/probes annealing to the nucleoprotein gene were used; forward primer: 5′CAG GAT CCC TTT GGC AGT TT, reverse primer: 5′ TAG GCT TCT CTT GCC CTT GT and probe: 6-carboxyfluorescein–5′-CCCATAAGGTCACCCTCTT-3′-6-carboxytetramethylrhodamine (Life Technologies, Carlsbad, CA, USA). Cycle threshold values, representing MARV genome copy numbers, were calculated and shown as genome equivalents (GE)/mL. To create a standard curve for reference, viral RNA extracted from the MARV challenged stock was diluted and similarly processed; genomic equivalent values were determined by plotting Ct values vs. pfu/mL. The lower limit of quantification (LLOQ) is 2.92 log_10_ GEq/mL and the upper limit of quantification (ULOQ) is 6.93 log_10_ GEq/mL.

#### 2.5.3. RT-qPCR for Quantification of VSV Viral RNA

VSV genomes to assess vaccinemia were quantified by RT-qPCR. Analytical qualification was performed and tested at Q2 Solutions, Durham, NC, USA (previously Focus Diagnostics, San Juan Capistrano, CA, USA). The assay was prospectively qualified and was deemed fit for its intended use. The assay targets RNA sequences that are unique to the VSV vector nucleoprotein (VSV-N). The results are quantitated using an external standard curve of rVSVΔG-ZEBOV-GP calibrators. To verify RNA extraction from the specimen and successful RT-PCR amplification, an internal control (MS2 RNA phage) is spiked into each sample prior to RNA extraction and is amplified in parallel with the rVSV N-N target with each specimen for the entire assay procedure. For dichotomous analyses, the determination of positive/negative is made relative to the lower limit of quantitation. Samples with a value greater than or equal to the limit of quantitation (≥800 copies/mL for plasma samples) are considered positive. Samples with a value less than the limit of quantitation are considered negative. For the calculation of geometric means, values less than the lower limit of quantitation (LLOQ) are replaced with half the LLOQ value (400 copies/mL). The reported value is used for ≥LLOQ and <ULOQ (upper limit of quantitation) results. For statistical analysis, the ULOQ value is used for results ≥ ULOQ.

Quality Control acceptance criteria are stated in the assay SOP. Each batch includes a high virus control, a low virus control, a negative control, and an extraction control. Each batch must pass all acceptance criteria listed for all controls. All assay control values are trended and monitored for excursions, and new critical reagents are qualified before use. The quantitative range is 800–1 × 10^7^ copies/mL.

### 2.6. Antibody Quantification

#### 2.6.1. Plaque Reduction Neutralization Assay for MARV

Neutralizing antibody quantification was assessed by PRNT. Briefly, sera were heat inactivated for 56 °C for 30 min. Sera were diluted two-fold in media and incubated with 100 pfu of MARV for 1 h at 37 °C. The virus–serum mixture was then added to a Vero cell monolayer, similar to the plaque assay protocol. After 10 days, plates were fixed and plaques visualized by crystal violet staining. Neutralization titers were based upon the average number of plaques in unneutralized wells (naïve serum) and calculated on a 50% cutoff for PRNT_50_ titer. Convalescent guinea pig serum from a previous study was used as a positive control. Individual samples were assayed twice (replicates) and their average was recorded as the titer. Reciprocal titers for the LLOQ = 20 and ULOQ = 640. For samples outside this range, the next 2-fold dilution was considered the titer; where <LLOQ, a value of 10 was used, and >ULOQ, a value of 1280 was used.

#### 2.6.2. ELISA for MARV-GP

Sera were tested in an IgG ELISA according to the Battelle procedure [30]. Briefly, sera, including positive and negative controls, were incubated on microtiter plates coated with purified MARV Angola GP. Plates were washed, and bound primary antibody from sera was amplified using a horseradish peroxidase-linked anti-human IgG secondary antibody and visualized by the addition of 3,3′,5,5′-tetramethylbenzidine substrate. Plates were read on an ELISA plate reader (BioTek, Winooski, VT, USA) at 450 nM; results are reported as ELISA units/mL.

#### 2.6.3. Pseudovirion Neutralization Assay (PsVNA)

Neutralizing antibodies were quantified by a fluorescent assay using a rVSV-ΔG-MARV-GP (Angola) pseudotyped luciferase-encoded virus [30,31]. Serially diluted heat inactivated serum was mixed with an equal volume of PsV and incubated for 60–75 min prior to transferring onto Vero E6 in opaque black-walled 96-welled plates. After another 60–75 min incubation, additional media were added and plates were incubated for 16–26 h. Cells were lysed with a lysis buffer (Promega, Madison, WI, USA, Cat No. E1941) and shook on a plate rocker at 200 revolutions/minute for 30–45 min. Relative light units (RLU) were measured using a luminometer (BioTek Synergy HTx).

### 2.7. CBC and Serum Chemistry

A CBC panel was analyzed on some days. Whole blood collected in EDTA tubes was analyzed on a HEMAVET VetScan MH5 machine (Drew Scientific, Dallas, TX, USA) according to manufacturer’s instructions. Individual NHP blood was run twice and the average value was recorded. Measurements included white blood cell (WBC), neutrophil (NEU), lymphocyte (LYM), monocyte (MON), eosinophil (EOS), basophil (BAS), red blood cell (RBC), platelet (PLT), total hemoglobin (HGB), hematocrit (HCT), mean cell volume (MCV), mean corpuscular hemoglobin (MCHC), mean platelet volume (MPV), plateletcrit (PCT), red cell distribution width (RDW), and platelet distribution width (PDWc, PDWs).

Alternations in serum chemistry was detected from the freshly acquired clarified serum from SST tubes, and run on either the Vetscan VS2 (Zoetis, Parsipanny, NJ, USA) or Piccolo Xpress (Abaxis, Union City, CA, USA) rotor, depending on the compound to be detected. Compounds detected in the Vetscan rotor included glucose (GLU), blood urea nitrogen (BUN), creatinine (CRE), calcium (Ca), albumin (ALB), total protein (TP), alanine aminotransferase (ALT), alkaline phosphatase (ALP), gamma glutamyltransferase (GGT), amylase (AMY), total bilirubin (TBIL), phosphate (Phos), sodium (Na), potassium (K) and globulin (GLOB). The Piccolo Xpress detects GLU, BUN, CRE, Ca, ALB, TP, ALT, ALP, GGT, AMY, uric acid (UA), aspartate aminotransferase (AST), and C reactive protein (CRP).

### 2.8. Gamma Irradiation

Select sera samples were inactivated by gamma irradiation (5Mrads) prior to removal from the BSL4. A positive control of EGFP-expressing EBOV, irradiated in the same batch, was titrated to confirm inactivation by both lack of EGFP expression and cytopathic effect in Vero cells infected for 10 days. A parallel non-irradiated EGFP-EBOV control was used as a comparison.

### 2.9. Statistical Analysis

#### 2.9.1. NHP Group Assignments

NHPs were assigned to either the 0.9% saline group (*n* = 3) or rVSV-MARV (*n* = 5) for both studies by a blinded statistician. First, NHPs were pair matched between males and females, then the pairs were randomly assigned to the vaccine or control group. Due to the unequal numbers between cohorts, lastly, one of the remaining male–female pairings was added to the vaccine group.

#### 2.9.2. Data Analysis

In general, two-tailed *t*-tests were used to compare data between groups on a specific day or within the same group to the baseline reading. A log-ranked (Mantel–Cox) test was used to analyze survival proportions. For large values (hematology: WBC, LYM, MON, NEU, EOS, BAS, RBS, PLT and viremia), data were log_10_-transformed prior to analysis. Log_10_-transformed viremia, vaccinemia, PRNT, hematology and serum chemistry data were analyzed using two-tailed unpaired *t*-tests. Weights were compared within each group by a two-tailed paired *t*-test to the day of either vaccination or challenge; comparisons between groups were not performed. ELISA and PsVNA data were analyzed using a one-tailed unpaired *t*-test; ELISA value of “0” was assigned as a “1” and PsVNA values less than the LLOQ were assigned a value of “10” for statistical analysis.

All figures were created in Prism 9.0 (GraphPad, Prism version 10). Tables and statistical analyses were performed in Excel (Microsoft, version 16.66.1 for most calculations). Unless otherwise noted, all summary statistics calculated geometric means and geometric standard deviations.

## 3. Results

### 3.1. rVSV-MARV Is Well Tolerated and Elicits a Strong Immune Response in Cynomolgus Macaques

#### 3.1.1. NHP Onboarding and Vaccination

Healthy, MARV-naive cynomolgus macaques that fit in the inclusion criteria were separated into two experiments based upon the dose of vaccine administered; Experiment 1 was conducted first with an IM-delivered single vaccine dose of 6.7 × 10^7^ pfu. All data were analyzed prior to the dose-down Experiment 2, which supplied a vaccine dose of 1.6 × 10^7^ pfu. Figure 1 shows the schematic diagram for the timing of vaccination and challenge. Blood draws are depicted in red boxes to quantify vaccinemia/viremia, CBC values, serum chemistry levels and antibody titers. NHPs were anesthetized and vaccinated on day 0. The vaccination was well tolerated as evidenced by consistent weights for all NHPs for 34 days post vaccination (Appendix A). For Experiment 2, the injection site was also monitored for a rash, but none was observed.

#### 3.1.2. rVSV-MARV Produces a Transient Vaccinemia

The amount of vaccine circulating through the blood, vaccinemia, is a critical component for vaccines, especially given environmental considerations like mosquito transmission or person-to-person spread [32]. Blood was taken from NHPs on days 1–4, 7, 14 and 34 (Figure 1), and clarified plasma was used to quantify rVSV-MARV levels by qRT-PCR. None of the 0.9% saline-treated NHPs showed any detectable levels of rVSV-MARV viral RNA. Conversely, all vaccinated NHPs were vaccinemic, peaking at 1 day post vaccination (dpv) and waning until all samples were at undetectable levels when tested on 14 and 34 dpv (Figure 2).

#### 3.1.3. Vaccination Elicits a Strong Antibody Response

A strong total antibody response is an important correlate of protection for Marburg virus infection [25]. Antibodies, both total anti-GP IgG and neutralizing, were quantified from blood collected prior to challenge and on days 7, 14, 34 and 42 post vaccination. Further, as NHPs were transported into the BSL4 for challenge, the effect of gamma irradiation to inactivate sera for use outside of the BSL4 was also investigated. First, anti-GP titers were quantified by ELISA assay (Figure 3).

As expected, there were no quantifiable serum IgG titers observed prior to vaccination and in the 0.9% saline-treated NHPs. Interestingly, similar to previous reports [33], there was a small effect of gamma irradiation on the ELISA titer in control sera; irradiation resulted in a slight increase (geometric mean +/− geometric standard deviation) in the signal on day 34, rising from undetectable to 31 +/− 1.2 and 11 +/− 1.1 ELISA Units/mL for Experiments 1 and 2, respectively. rVSV-MARV vaccinated NHPs began to show anti-GP antibodies by 14 dpv (92.7 +/− 3.5 and 216.0 +/− 1.8 Units/mL in Experiments 1 and 2, respectively), which increased by 34 dpv (601.6 +/− 2.7 and 785.8 +/− 2.1 Units/mL in Experiments 1 and 2, respectively). Unlike the controls, irradiation appeared to slightly reduce ELISA titer on day 34 (544.3 +/− 2.3 and 577.8 +/− 2.1 Units/mL in Experiments 1 and 2, respectively). All NHPs had binding antibody responses at the time of MARV challenge at day 42 (550.6 +/− 2.7 and 588.5 +/− 2.4 Units/mL in Experiments 1 and 2, respectively).

Neutralizing antibodies provide another metric for analyzing vaccine efficacy. Here, both PsVNA and PRNT assays quantify the levels of neutralizing antibodies. PsVNA titers are shown in (Figure 4). As expected, no detectable titers were measured prior to vaccination or in the 0.9% saline-treated sera, and values remained <LLOQ regardless of irradiation status. Similar to the effect of irradiation on ELISA titer, there was a reduction in sensitivity on PsVNA_50_ titer; one sample was slightly reduced from a detectable PsVNA_50_ titer of 23.55 to below the LLOQ of 20 [34]. Geometric mean titers on day 34 ranged between non-irradiated and irradiated samples from 77.1 +/− 2.1 and 64.4 +/− 3.2 for Experiment 1 and 192.2 +/− 2.2 and 56.1 +/− 1.8 for Experiment 2. Regardless, by 34 dpv, all NHPs had detectable PsVNA_50_ titers, which increased slightly by the day of challenge (42 dpv, irradiated titers of 79.4 +/− 1.6 and 129.3 +/− 1.6 for Experiments 1 and 2, respectively).

Neutralizing antibodies were also quantified by PRNT assay against MARV Angola virus (Figure 5). Congruent with the PsVNA results, none of the NHPs prior to vaccination or treated with 0.9% saline had a positive PRNT_50_ titer (defined as a reciprocal average titer of two replicates > 20). In Experiment 1 and 2, 4/5 rVSV-MARV-vaccinated NHPs in each experiment had at least one replicate titration with a detectable titer. Titers were first noted on 14 dpv. The titers between 0.9% saline and rVSV-MARV were not statistically significant on days 14, 34 or 42 for Experiment 1 (*p* = 0.08, 0.16 and 0.37, respectively) or on day 14 or 34 for Experiment 2 (*p* = 0.17, 0.13, respectively) except on day 42, where the difference was significant (*p* = 0.043). After challenge, there was an increase in PRNT_50_ titer, consistent with an anamnestic response. Interestingly, one 0.9% saline-treated NHP (MB105) had a detectable reciprocal PRNT_50_ titer of 20 at the terminal bleed (8 dpc).

### 3.2. rVSV-MARV Fully Protects NHPs from Lethal MARV Infection

All NHPs were challenged intramuscularly with MARV Angola at 1 *×* 10^4^ pfu, a uniformly lethal dose. Clinical scores were taken daily until the end of the study and blood samples were taken for viremia clinical chemistry and hematology. Temperature was not taken nor fever quantified.

#### 3.2.1. Survival

MARV, at the dose given, is a lethal infection with a predictable disease progression (Figure 6). No clinical scores or morbidity were recorded for rVSV-MARV vaccinated NHPs in either Experiment 1 or 2. The 0.9% saline-treated NHPs, however, started to show signs of loss of appetite, decreased activity and petechiation/ecchymosis on day 6 post challenge. Signs increased in severity until reaching morbidity criteria requiring euthanasia on days 7 and 8. One NHP in the Experiment 1, 0.9% saline-treated group, did not meet euthanasia criteria on the last check on day 7 but succumbed to infection prior to the first check the next morning.

#### 3.2.2. Viremia

MARV viral RNA levels were assessed in sera samples taken at the time of challenge and at days 5, 7, 10, 14 and 28 and at the time of euthanasia/death (Figure 7). No viremia was detected in any rVSV-MARV-vaccinated NHP. However, one NHP, UG908, had an initial titer of 3.44 log_10_ GE/mL on day 14 post challenge (0.5 mL serum tested). When a frozen aliquot (0.35 mL tested) was re-extracted and quantified, viremia was undetected (<LLOQ), and this value is shown in Figure 7. Peak viremia, observed in 0.9% saline-treated NHPs, was detected between 5.4–5.69 log_10_ GEq/mL serum in Experiments 1 and 2, respectively. Interestingly, the peak day of viremia for Experiment 1, 5 dpc, was the lowest viremia for Experiment 2, despite a nearly identical challenge dose of ~1 × 10^4^ pfu/NHP. Sera from NHPs were collected upon necropsy on 8 dpc but no sample for NV603 (Experiment 2) was obtained.

### 3.3. Hematology

Blood was collected from NHPs on 0, 5, 7, 10 and 28 dpc and at/around the time of death (morbidity, pooled 7 dpc and 8 dpc) for blood cell composition and serum chemistry analyses. Individual blood samples were run twice for CBC analysis, and the average of these two values was recorded. Fresh (non-frozen) sera were analyzed using the Vetscan or Piccolo Rotors, depending on the assay required; for example, C-reactive protein is part of the Piccolo panel, but not the Vetscan panel. Baseline data on 0 dpc, obtained from both Vetscan and Piccolo rotors, were similar. Factors associated with clotting, like prothrombin time, were not measured.

#### 3.3.1. CBC

CBC values for NHPs are summarized in Figure 8. Additional blood samples were taken immediately after death; samples were analyzed either on day 7 or 8 and combined together under “moribund”. However, only 2 moribund samples for each Experiment were obtained so no statistics were performed.

WBC (Figure 8A): Consistent with previous MARV NHP models [12,13], total WBC values increased as MHF progressed, but only for the unvaccinated NHPs, which trended towards significance. Prior to the moribund state and compared to the rVSV-MARV-treated NHPs, control animals were mildly leukopenic on 5 dpc and beginning to show leukocytosis by day 7–8, when they succumbed to the MARV challenge. However, around the time of death, WBC values increased. There were no differences between Experiments 1 and 2 for either 0.9% saline-treated or rVSV-MARV vaccinated NHPs on the day of challenge. When the unvaccinated vs. vaccinated values were compared within each experiment on each day, the only difference was seen in Experiment 1 on 5 dpc (two-tailed unpaired *t*-test, *p* = 0.023).

LYM (Figure 8B): Similar to WBC counts, lymphocytes were also reduced during the intermediate phase of infection (5 dpc) for both studies. The difference between the 0.9% saline-treated and vaccinated NHPs did not achieve significance for Experiment 1 (*p* = 0.063), but it did for Experiment 2 (*p* = 0.008), consistent with previous reports [12,13]. Unlike WBC counts, LYM levels were not elevated at morbidity. Lastly, rVSV-MARV vaccinated NHPs showed no statistical changes in LYM during the challenge phase.

MON, NEU, EOS and BAS (Figure 8C–F): There were no significant alterations in monocyte, neutrophil, eosinophil or basophil levels during the infection for both studies. In some cases, levels rise as the disease becomes more severe.

RBC and HBG (Figure 8G,I): Except for Experiment 2 at the time of challenge, there were no differences in red blood cell [RBC] or hemoglobin [HBG] levels between the 0.9% saline and rVSV-MARV groups on any day, even at the time of morbidity/death, where significant signs of petechia and hemorrhage were observed in the non-vaccine protected groups.

PLT (Figure 8H): The only significant changes between the 0.9% saline and vaccinated NHPs were observed at the time of challenge (Experiments 1 and 2, *p* = 0.04, *p* = 0.03, respectively). One datapoint was omitted from analysis due to a bad reading (NHP UG883 Experiment 1 on day 5, reading of “0” on both runs). After challenge, there were no statistical differences between the treatments in platelet levels for both studies.

#### 3.3.2. Serum Chemistry

Serum clarified from blood was analyzed for changes in key ion and protein levels on days 0 (baseline), 5, 7, 10 and 28. Additional sera was obtained peri- and post-mortem; blood obtained from one NHP in Experiment 2 on day 7 (NV603) was clotted and not suitable for analysis. Sera were run on the Piccolo rotors on days 0 and 7 (and day 10 of Experiment 2 only) to obtain additional readings for C reactive protein (CRP) and AST markers, which are absent on the Vetscan rotors but were shown to be elevated in the MARV-infected patients [35].

CRP: High levels of serum C reactive protein [CRP] result from liver inflammation and infection, and are a marker of filovirus infection [12,36]. CRP levels were all less than 5 mg/L (assay LLOQ) on day 0, except for one NHP in the rVSV-MARV-vaccinated group from Experiment 1 (UG908) which was 8.2 mg/L, which is slightly above the normal levels of 0–7.5 mg/L. Sera were tested again on 7 dpc, when there was significant MHF in the 0.9% saline group. CRP levels could not be quantified for 2 NHPs in Experiment 1, as the reading was outside the assay’s range (>ULOQ = 200 mg/L). The remainder of the 0.9% saline-treated NHPs had levels above 26 mg/L (Table 1). The majority (7 of 10) of the rVSV-MARV vaccinated NHPs had undetectable CRP (<5 mg/L) levels, (excluding UG908, which had elevated levels post vaccination and pre challenge). Two rVSV-MARV-vaccinated NHPs had quantifiable values within the normal range of 5.3 and 6.3 mg/L, but one (UG1044) had an elevated reading of 126 mg/L. Values on day 10 (*n* = 5) and 28 (*n* = 4) dpc for rVSV-MARV vaccinated NHPs in Experiment 2 were all <5 mg/L, with the exception of UG1037 with a value of 5.1 mg/L on 28 dpc.

##### Liver Enzymes and Metabolites

AST (Figure 9A): Elevated levels of AST are associated with fatal human EBOV [37] and the macaque MARV model (reviewed in [38]). Serum AST levels were obtained on days 0 and 7 post challenge (Piccolo rotor). On the day of challenge, the geometric mean for any group ranged between 26.83 U/L +/− 1.15 to 32.86 U/L +/−1.11. By day 7, the average titers in the rVSV-MARV groups increased slightly to 36.61 U/L +/− 1.22 and 42.12 U/L +/− 1.22, Experiments 1 and 2, respectively. In the 0.9% saline treated NHPs, however, many of the values exceeded the ULOQ of 2000, and one NHP in Experiment 2 had an uninterpretable reading (NV603). Although proper statistics cannot be performed, significance can be inferred if the values of 2001 U/L are given to the NHPs with values exceeding the ULOQ; unvaccinated NHPs had statistically higher AST levels compared to the vaccinated groups on the same day (*p* < 0.001).

ALT (Figure 9B,C). Along with AST, alanine aminotransferase levels are correlated with filovirus disease [13,38,39]. ALT readings were obtained from both the Vetscan and Piccolo rotors. On the day of challenge, the geometric mean ALT levels for different treatment groups ranged between 25.46–55.29 U/L +/− 1.78–1.47. Statistically significant differences between 0.9% saline and rVSV-MARV groups were observed on days 5 (*p* = 0.036 and *p* < 0.01 for Experiments 1 and 2, respectively) and 7 (*p* < 0.001 both Experiments) post challenge. NHP NV603 (Experiment 2, 0.9% saline group) had an ALT level exceeding the ULOQ of 2000 U/L on 7 dpc, so the data for quantification were assigned to be 2001 U/L. The highest levels were seen on days 7 and at the time of morbidity (NHP dependent, at 7 or 8 dpc). The levels of ALT remained low for the rVSV-MARV groups until the end of the study at 28 dpc.

ALP (Figure 9E,F) High alkaline phosphatase levels are usually indicative of liver, kidney and/or bone disease. ALP levels in 0.9% saline groups increased with MHF severity. By 5 dpc, the difference between unvaccinated and vaccinated groups in both Experiments was significant (*p* = 0.011 Experiment 1 and 0.015 Experiment 2), which increased in significance by 7 dpc (*p* < 0.001) in both Experiments. The ALP levels in rVSV-MARV vaccinated NHPs did not alter significantly after MARV infection.

GGT (Figure 9D): High levels of serum gamma glutamyltransferase levels can be a sign of liver and bile duct damage. GGT levels were detected only from the Piccolo rotor and assessed on 7 dpc. However, while there was no difference between the unvaccinated and vaccinated groups at baseline, the GGT levels were significantly higher in 0.9% saline-treated NHPs on 7 dpc for both Experiments (*p* < 0.001).

TBIL (Figure 9G): High total bilirubin results from when the liver cannot process bilirubin or the bile ducts are blocked. No significant changes were observed in either Experiment between 0.9% saline and rVSV-MARV groups at baseline or 5 dpc, but the levels were much higher at the time of morbid MHF.

##### Renal Metabolites

Patients with MHF, especially those with severe illness, often have impaired renal function (reviewed in [40]). Creatinine (CRE) and blood urea nitrogen (BUN) tests are elevated in fatal filovirus-infected patients [39] and MARV-infected cynomolgus macaque models [12,13].

CRE (Figure 10A,B). CRE levels were not different between 0.9% saline and rVSV-MARV groups at the start of the study or on 5 dpc. However, CRE levels were significantly increased on 7 dpc in control animals (*p* = 0.02 in Experiment 1 and <0.001 Experiment 2). No statistics were run for the samples at morbidity; however, the levels were approximately doubled at 7 dpc.

BUN (Figure 10C,D). BUN levels showed a yo-yo like trend; the 0.9% saline groups in Experiments 1 and 2 were significantly lower (*p* = 0.029 Experiment 1 and 0.040 Experiment 2) than rVSV-MARV on 5 dpc. By 7 dpc, only Experiment 2 showed a difference between these groups (*p* < 0.001) and BUN reached their highest levels at morbidity.

##### Other Significant Metabolites and Electrolytes

Additional serum chemistries with alterations during MHF are shown in Figure 9.

GLU (Figure 11A,B). Non-fasting glucose levels were examined on both Vetscan and Piccolo rotors. The 0.9% saline and vaccinated groups were statistically different in Experiment 2 on the day of challenge, but no differences were observed on 5 dpc. Rather, the values in non-vaccinated NHPs were much lower as the disease progressed on 7 dpc (*p* = 0.011) and at the time of morbidity.

Ca (Figure 11C,D). Hypocalcemia is associated with a variety of health issues. Calcium levels were significantly lower on day 5 (Experiment 1, *p* < 0.01, Experiment 2, *p* = 0.027) and 7 dpc (*p* < 0.01) in the 0.9% saline-treated NHPs for both experiments.

TP (Figure 11E,F). Decreased serum total protein is associated with kidney and liver damage and affects fluid balances. Hypoproteinemia was observed on 7 dpc in 0.9% saline-treated NHPs in Experiments 1 and 2 (*p* = 0.49 and *p* = 0.003, respectively).

AMY (Figure 11G,H). Low serum amylase levels could suggest pancreas, liver or kidney disease. There were no differences between the unvaccinated and vaccinated groups or Experiments at baseline. However, AMY levels in the unvaccinated groups decreased significantly, only in Experiment 2, by 5 dpc (*p* = 0.02) and 7 dpc (*p* = 0.006). Experiment 1 trended towards significance on 7 dpc (*p* = 0.054).

ALB (Figure 11I,J). Low levels of serum albumin can be signs of kidney or liver disease. No significant differences were noted between groups in either experiment on any day compared, but levels trended down at the time of morbidity.

### 3.4. Necropsy Findings

Necropsies were performed on all NHPs as close to the time of death as possible (on days 7 and 8 post challenge) and at the end of the study. Reports were prepared by board-certified veterinarians with extensive experience with filovirus pathology in macaques. Hallmark pathology associated with MHF was noted in all 0.9% saline-treated NHPs. Briefly, petechiation was observed on the trunk and limbs. Evidence of hemorrhage was observed in all, including coffee ground-like contents in the stomach, red pinpoint discoloration of the mucosa in the intestines, friable and pale yellow livers, and discoloration in target organs, including the adrenal glands, rectum, testes, kidneys and lungs. The brain was not observed due to the difficulty in accessing this organ in containment.

All rVSV-MARV vaccinated NHPs had unremarkable pathology, apart from one animal (UG908) in Experiment 1; lung lobe focal adhesions were noted on 28 dpc. However, no tissues were collected for further analysis so was not possible to determine the cause or if this was a pre-existing condition prior to vaccination and/or challenge.

## 4. Discussion

The purpose of these experiments was to expand upon the foundational data obtained from rVSV-MARV-vaccinated guinea pigs in the well-characterized cynomolgus macaque model [26]. A single dose of rVSV-MARV (PHV01) protected NHPs against a lethal MARV infection. No mortality or adverse clinical signs, including rash in Experiment 2, or changes in weight were observed post vaccination, similar to other studies evaluating VSV-vectored vaccines [16]. Antibody responses, both IgG and neutralizing, as determined by fit-for-purpose assays, were established within 14 days. Vaccinemia was noted and resolved by day 14 post vaccination and is consistent with other rVSV vaccines; Jones et al. described low levels of rVSV RNA detected in 4 of the 6 tested animals at 2 days post vaccination by RT-PCR [8]. These data are important, as wild type VSV is transmitted by hematophagous flies and *Culicoides* midges [41], and a critical feature of an arthropod-transmitted virus used as a vaccine vector is the inability to be transmitted by blood-feeding insects. Interestingly, 100% of the patients administered the rVSVΔG-ZEBOV-GP vaccine also produced a vaccinemia in clinical trials (summarized in [32]). However, this report concluded that the environmental risk of rVSVΔG-vectored filovirus vaccines is low, as the vector is significantly attenuated and does not replicate in these transmission vectors [32].

Antibodies are believed to be essential for rVSV vaccine-mediated protection in the case of ZEBOV, and the same is likely to be the case for MARV, but a seroprotective level of antibody has not yet been demonstrated, and the functional role of antibodies (e.g., neutralization, Fc-mediated cell targeting, or both) is not defined. The antibody response is believed to be responsible for protective immunity, as shown by the protection through the administration of both polyclonal and monoclonal antibodies against MARV in animal models [42]. In fact, Dye et al. demonstrated that passively transferred species-specific anti-MARV IgG, administered 48 h post lethal challenge followed by additional treatments, completely protected rhesus macaques from death and disease [42,43]. In our study, anti-MARV GP binding antibodies were not detected in control animals, but MARV GP binding antibodies were detected in all rVSV-MARV-vaccinated NHPs. This typical IgG production upon vaccination was consistent with other rVSV-MARV candidates [8,16,17,19,22,24,44,45]. The production of specific anti-MARV GP antibodies has been detected at various time points post vaccination in these studies, including one study that demonstrated antibody production as early as 10 days post vaccination [24]. Interestingly, the induction of a long-lasting humoral response was also evidenced by the detection of anti-MARV GP IgG as late as a year after immunization [16].

MARV-specific neutralizing antibodies were detected in 7 of the 10 rVSV-MARV vaccinated NHPs prior to MARV challenge by PRNT_50_, but in all NHPs by PsVNA assay. Importantly, neutralizing antibodies (PsVNA) were detected as early as 7 days post-vaccination in 20% of vaccinated NHPs. A neutralizing titer of 1:20 has been shown to be protective post-exposure in previous reports [16,19,25]. In fact, it has been shown that while titers can fluctuate to undetectable levels over time, vaccinated animals can still be protected from MARV challenge, as shown by Mire et al. [16]. Jones et al. observed similar results, with only two out of six vaccinated Cynomolgus macaques presenting detectable neutralizing antibody titers against MARV on the day of challenge [8]. In another study, surviving MARV-infected Rhesus macaques treated with a vaccine shortly after challenge showed only low levels of neutralizing antibodies, which were first detected after 10 days post challenge [25]. Importantly, O’Donnell et al. showed that complete protection from disease can be achieved even when antibody neutralizing titers are not different between vaccinated and unvaccinated animals at the time of challenge [45]. Overall, a protective immune response may be associated with additional components of the immune system other than neutralizing antibodies, such as non-neutralizing antibodies [8,25]. Assessing the immune response induced by our vaccine candidate in greater detail will be the scope of future studies.

All rVSV-MARV-vaccinated NHPs survived MARV challenge without any notable clinical signs of MARV disease, and viremia was below the LLOQ for all but one observation day. This is a remarkable level of protection from death and viremia, considering the very high MARV challenge dose (~10,000 × LD_50_)_._ All the 0.9% saline-treated NHPs succumbed to MARV challenge. Other rVSV-MARV vaccine candidates have shown similar results, where all immunized cynomolgus macaques were protected from lethal MARV challenge 28 days [8,19,22,44], 35 days [17], or even a year [16] after vaccination, with no disease signs or detectable viremia. Even with short periods of time between vaccination and challenge, rVSV vaccines have shown high efficacy. Complete survival was observed with challenge 14 [45,46] or 7 days post vaccination. Other cohorts in some of these studies had 80% or 75% survival when the vaccine was administered 5 [47] or 3 [46] days prior to challenge, respectively [46]. rVSV-MARV vaccine candidates have also shown protective efficacy when administered after challenge. High survival was observed in one study, with cynomolgus macaques treated 24 h (83% survival) or 48 h (66% survival) after challenge with the MARV Musoke strain [20]. A different study, with Rhesus macaques treated with different rVSV-MARV vaccines 20–30 min after a low dose, but a lethal challenge of MARV, also showed high survival rates (80–89%) [25].

Many of the parameters observed in this study are consistent with the natural history studies published by others [13]. In these studies, the challenge dose achieved aligns most closely with Comer et al. [12]. Upon MARV challenge, the virus can be detected not only in the serum of unvaccinated cynomolgus macaques, but also in several organs, including liver, spleen, lung, kidney, pancreas, testes, brain, bone marrow, among others [21]. The CBC values obtained in these studies were mostly aligned with filovirus clinical findings and similar to natural history studies. Leukopenia, during the intermediate phase of MHF observed here, is consistent with the observations in human [48] and NHP models [12,13]. No monocyte alternations were observed in natural history studies, but these studies noted neutrophilia on 4, 7 and 9 dpc [13] or 3, 5 and 7 dpc [12]. There is a little discrepancy in PLT results, as Comer et al. reports a decrease from baseline or for uninfected controls in platelet counts [12], whereas Zumbrun et al. does not report a similar trend [13].

Liver and renal enzymes were significantly altered in unvaccinated NHPs. AST and ALT, specifically, are correlated with severe or fatal filovirus disease in humans [39,40]. These were also significantly elevated, especially around peak disease and death time, in the cynomolgus macaque model [12,13]. Similar to other liver enzymes, GGT is elevated in severe MHF in NHPs [13], guinea pigs [49] and rhesus macaques [35]. TBIL is elevated in rhesus [35] and cynomolgus macaques [13,22,38], and MARV infection models at late stage MHF. Kidney function protein CRE and metabolite BUN are significantly elevated in Ebola patients [39]. Alterations in these markers correlate well with the gross pathology observed in the liver and kidneys at necropsy in the 0.9% saline-treated NHPs.

Marzi and collaborators observed in two studies that unvaccinated animals infected with MARV Angola had increased levels of AST [46] and ALT [46], as well as important inflammatory mediators, such as IL-1β, IL-6, IL-1Ra, MIP-1α, TNF, IFN-γ, IL-2, and FGF-β [17]. They also noted a trend of increases in white blood cell and lymphocyte numbers, though it was not statistically significant [17]. Similar to our findings, studies with rhesus [25] and cynomolgus [47] macaques found that lethal MARV infection led to increased levels of ALT, AST, ALP, GGT, BUN, CRE, and C-reactive protein. In line with our results, these studies also found hemorrhagic manifestations and other hematological alterations, such as lymphopenia, thrombocytopenia, and neutrophilia in lethal cases [47].

rVSV-MARV-vaccinated NHPs showed no clinical signs of illness and had only minor alterations in CBC or serum chemistry values after MARV challenge (unless otherwise noted above). The trends between Experiments 1 and 2 were similar. The most prominent differences observed in the 0.9% saline-treated NHPs were observed for the liver enzymes ALT, AST, ALP and GGT, and TBIL at the time of morbidity, which remained unaltered post challenge in rVSV-MARV-treated NHPs. Unfortunately, CRP would not be fully quantified due to issues with the assays. In Experiment 1, two readings were outside the assay’s linear rage (>ULOQ of 200 mg/L). With one exception, all rVSV-MARV-vaccinated NHPs had readings below the normal range of 7.5 mg/L, which was NHP UG1044 with a 7 dpc reading of 126 mg/L. Interestingly, other liver enzyme values for this NHP at this time were not elevated.

The gross pathology described in unprotected 0.9% saline-treated NHPs at the time of morbidity or death was consistent with lethal MARV infection, including hemorrhage in liver, GI tract and kidneys. The brain was not observed due to difficulty and safety concerns. Upon the end-of-study necropsy of the surviving rVSV-MARV-vaccinated NHPs, one NHP in Experiment 1 was found to have right caudal lung lobe focal adhesion to diaphragm. No samples were taken for viral load analysis or pathology, so it was impossible to determine the cause. During the study, this NHP had anti-MARV titers (ELISA, PsVNA and PRNT_50_). Interestingly, MARV RNA was detected in this NHP on day 14 post challenge (Ct value 35.96, corresponding to 3.44 log_10_ GE/mL in 0.5 mL serum). When the assay was repeated with a frozen serum aliquot (0.35 mL, re-extracted RNA), viremia was below the assay LLOQ (Table 1). No CBCs or serum chemistry were taken at this timepoint, so correlations cannot be made.

These data provide very encouraging results, warranting the continued study of the rVSV-MARV PHV01 vaccine. This vaccine has the potential to fill a gap in much-needed countermeasures for safe and effective vaccines to protect against lethal MHF.

## Figures and Tables

**Figure 1 viruses-16-01181-f001:**
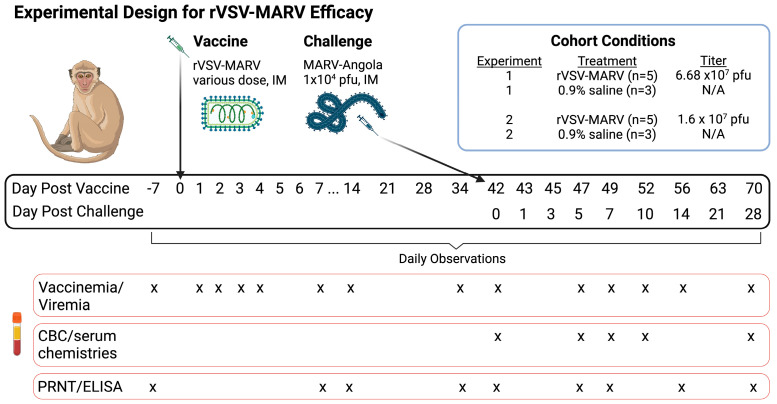
Schematic Diagram of rVSV-MARV Efficacy Testing. Cynomolgus macaques were divided into two experiments based upon the titer of the vaccine. Experiments 1 and 2 follow the same schedule, blood draws are shown in red boxes. Image made using BioRender.com (accessed on 1 June 2024).

**Figure 2 viruses-16-01181-f002:**
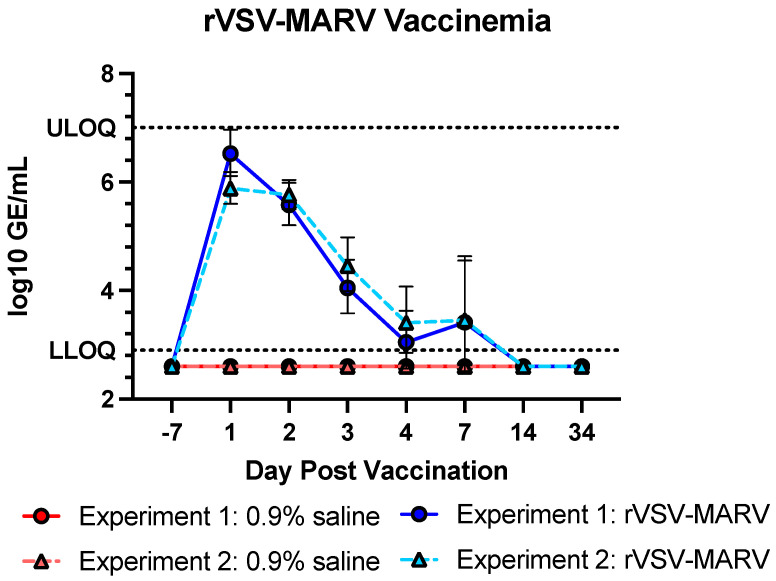
Vaccinemia Post Vaccination. Group vaccinemia titers (geometric mean) obtained from plasma are shown; red/pink points show 0.9% saline-treated NHPs, and blue/light blue points show rVSV-MARV-vaccinated NHPs. Solid lines with circle points denote Experiment 1, dashed line with triangle points denotes Experiment 2. Values represent geometric means, error bars denote geometric standard deviation. Sample values calculated > ULOQ or <LLOQ were shown as double or half these amounts, respectively. Data from 0.9% saline Experiments 1 and 2 are superimposed on the graph. LLOD (lower dotted line) = 2.9 log_10_ GE/mL, ULOQ = 7 log_10_ GE/mL (upper dotted line).

**Figure 3 viruses-16-01181-f003:**
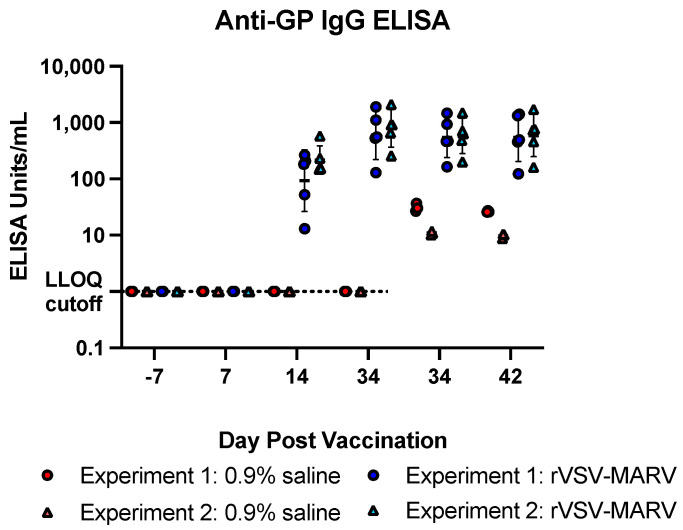
ELISA IgG titers post vaccination. Anti-MARV GP IgG ELISA titers for individual NHPs are shown. Red/pink filled icons show 0.9% saline-treated NHPs, blue/light blue filled icons show rVSV-MARV vaccinated NHPs. Closed circles show data from Experiment 1 and triangles show Experiment 2. Horizonal lines in a dataset show geometric mean and error bars denote geometric standard deviation. Any ELISA value of 0 was set to 1, and the LLOQ is shown by the dashed line (non-irradiated samples).

**Figure 4 viruses-16-01181-f004:**
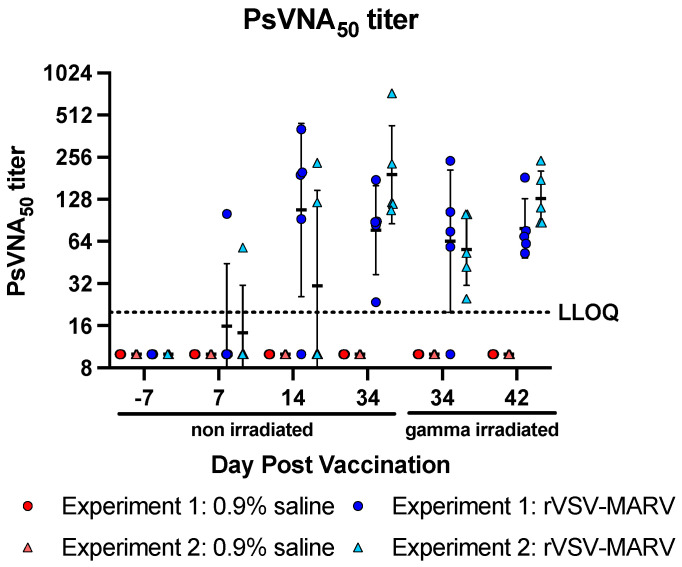
Neutralization titers post vaccination (PsVNA). PsVNA titers for individual NHPs are shown. Red/pink icons show 0.9% saline-treated NHPs, blue/light blu e icons show rVSV-MARV vaccinated NHPs. Closed circles show data from Experiment 1 and triangles show Experiment 2. Geometric means are displayed as horizontal bars and error bars denote geometric standard deviation. The horizontal dashed line shows the LLOQ of the assay of 20.

**Figure 5 viruses-16-01181-f005:**
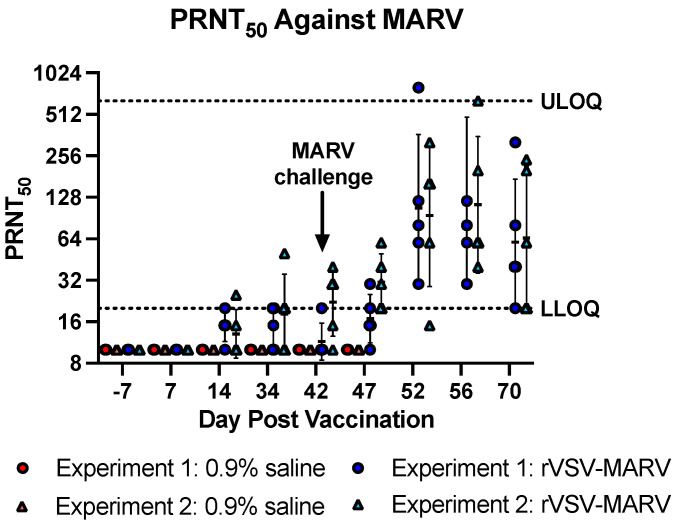
Neutralization titers post vaccination (PRNT_50_). PRNT_50_ titers for averaged individual NHPs are shown. Red/pink icons show 0.9% saline-treated NHPs, blue/light blue icons show rVSV-MARV vaccinated NHPs. Closed circles show data from Experiment 1 and triangles show Experiment 2. Error bars denote geometric standard deviation. The horizontal dashed line shows the LLOQ of the assay of 20 and ULOQ of 640. Undetectable values are listed as ½ LLOQ = 10. Some sera have titers of 15, where one replicate was undetectable and the other was 20.

**Figure 6 viruses-16-01181-f006:**
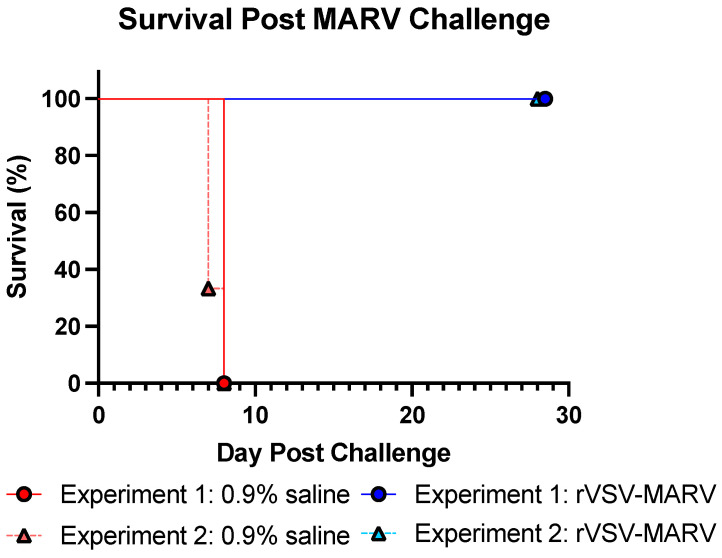
Survival post MARV challenge. Group survival from challenge with 1 × 10^4^ pfu MARV; red/pink points show 0.9% saline-treated NHPs, and blue/light blue points show rVSV-MARV-vaccinated NHPs. Solid lines denote Experiment 1; the dashed line denotes Experiment 2. Comparisons between 0.9% saline and rVSV-MARV survival curves within each experiment were significant (Log-rank test, *p* = 0.008, 0.004 for Experiments 1 and 2, respectively).

**Figure 7 viruses-16-01181-f007:**
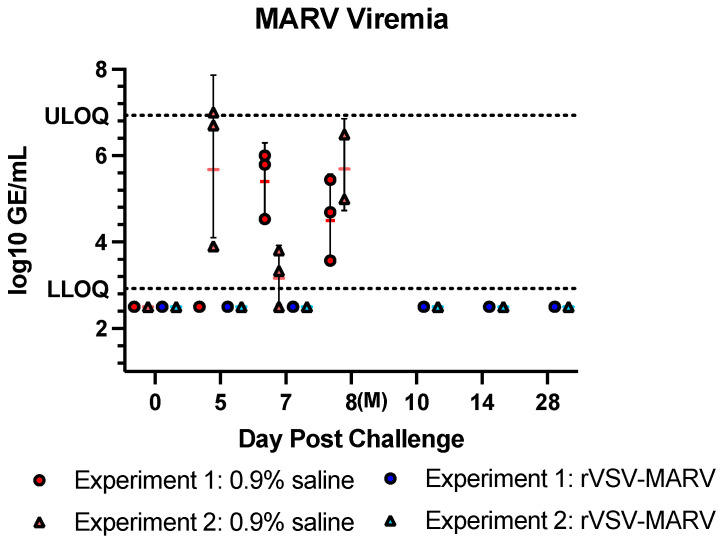
MARV viremia post challenge. Sera taken at various timepoints challenge were assessed by qRT-PCR. Red/pink points show 0.9% saline-treated NHPs, and blue/light blue points show rVSV-MARV-vaccinated NHPs. Only the 0.9% saline-treated NHPs were bled at a terminal timepoint on 8 dpc. Error bars denote geometric standard deviation. The horizontal dashed line shows the LLOQ of the assay of 2.92 and ULOQ of 6.93. Undetectable values are listed as 2.5.

**Figure 8 viruses-16-01181-f008:**
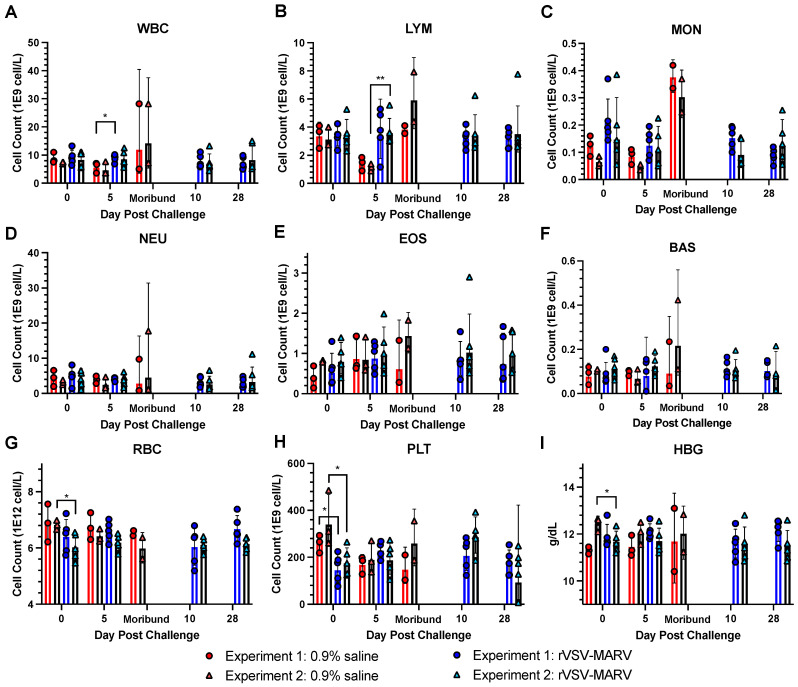
CBC alterations post MARV challenge. Values from individual NHPs for (**A**) WBC, (**B**) LYM, (**C**) MON, (**D**) NEU, (**E**) EOS, (**F**) BAS, (**G**) RBC, (**H**) PLT and (**I**) (HBG). Red/pink circles show 0.9% saline-treated NHPs, and blue/light blue triangles show rVSV-MARV-vaccinated NHPs. Solid red or blue bars denote Experiment 1, black bars Experiment 2, where the height is equal to the geometric mean of the group. Error bars denote geometric standard deviation. A two-tailed unpaired *t*-test compared 0.9% saline and rVSV-MARV groups on a specific day (* *p* < 0.05, ** *p* < 0.01).

**Figure 9 viruses-16-01181-f009:**
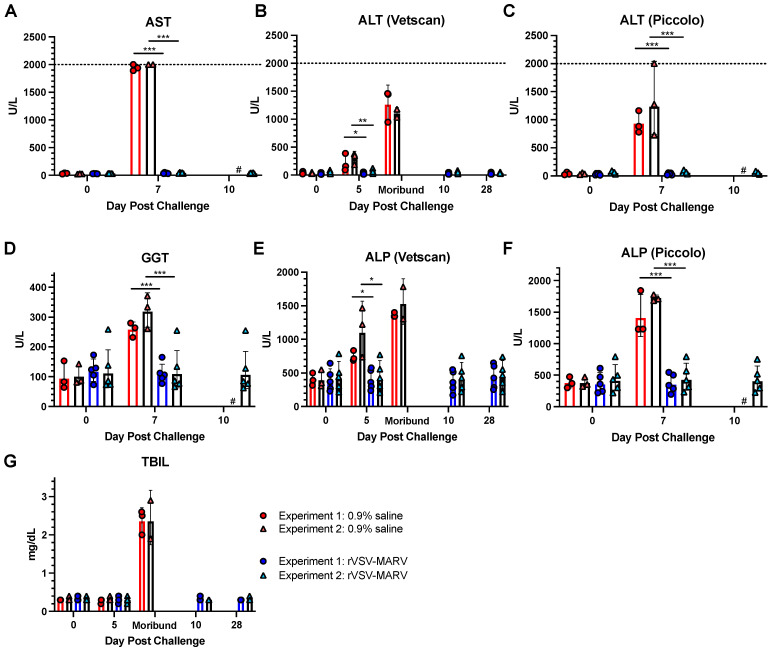
Liver enzyme and metabolite values post MARV challenge. Levels of (**A**) AST on the Piccolo rotor, (**B**) ALT on the Vetscan rotor and (**C**) ALT on the Piccolo rotor, (**D**) GGT on the Piccolo rotor, (**E**) ALP on the Vetscan rotor, (**F**) ALP on the Vetscan Piccolo, (**G**) TBIL on the Vetscan rotor. Red/pink circles show 0.9% saline-treated NHPs, and blue/light blue triangles show rVSV-MARV-vaccinated NHPs. Solid red or blue bars denote Experiment 1, black bars Experiment 2, where the height is equal to the geometric mean of the group. Error bars denote geometric standard deviation. The ULOQ is depicted as a horizontal dashed line. Any values < ULOQ were assigned a value of 2001 for analysis. Day 10 pc Piccolo values are only available for Experiment 2 (unavailable cohort data shown as #). A two-tailed unpaired *t*-test compared the 0.9% saline and rVSV-MARV groups on a specific day (* *p* < 0.05, ** *p* < 0.01, *** *p* < 0.001).

**Figure 10 viruses-16-01181-f010:**
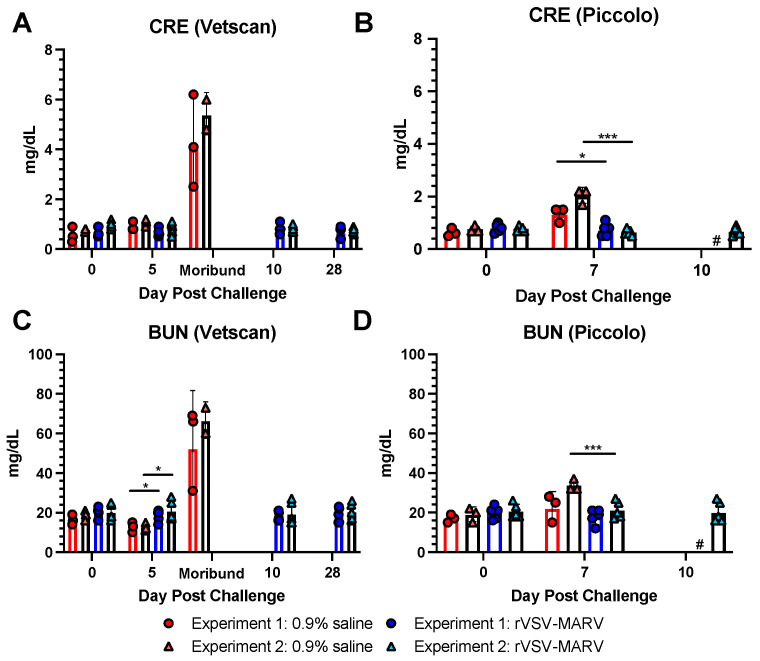
Renal metabolite values post MARV challenge. Levels of (**A**) CRE on the Vetscan rotor, (**B**) CRE on the Piccolo rotor, (**C**) BUN on the Vetscan rotor and (**D**) BUN on the Piccolo. Red/pink circles show 0.9% saline-treated NHPs, and blue/light blue triangles show rVSV-MARV-vaccinated NHPs. Solid red or blue bars denote Experiment 1, black bars Experiment 2, where the height is equal to the geometric mean of the group. Error bars denote geometric standard deviation. Day 10 pc Piccolo values are only available for Experiment 2 (unavailable cohort data shown as #). A two-tailed unpaired *t*-test compared the 0.9% saline and rVSV-MARV groups on a specific day (* *p* < 0.05, *** *p* < 0.001).

**Figure 11 viruses-16-01181-f011:**
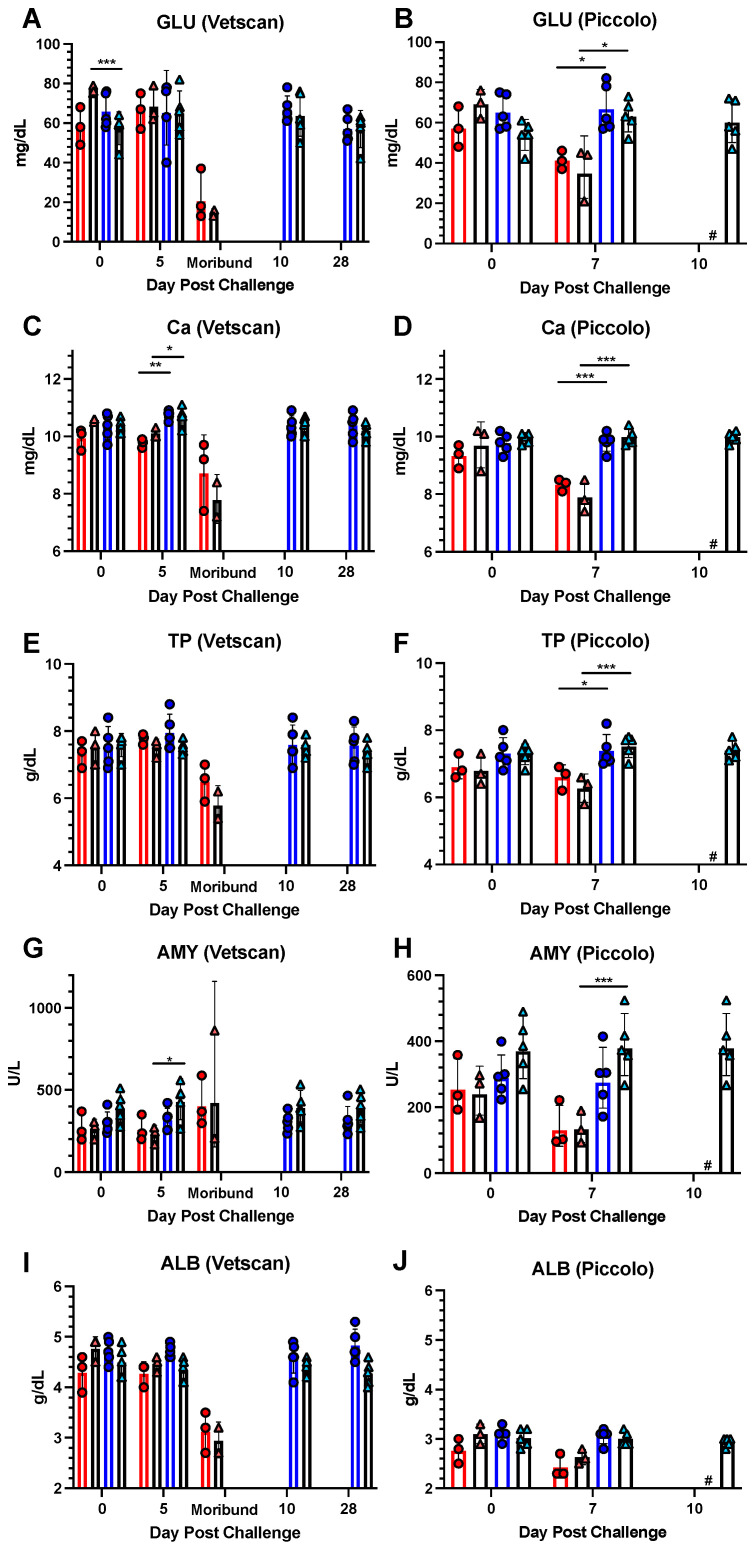
Other significant metabolite and electrolyte values post MARV challenge. Levels of glucose [GLU] measured on the (**A**) Vetscan and (**B**) Piccolo rotors, calcium [Ca] measured on the (**C**) Vetscan and (**D**) Piccolo rotors, total protein [TP] measured on the (**E**) Vetscan and (**F**) Piccolo rotors, amylase [AMY] measured on the (**G**) Vetscan and (**H**) Piccolo rotors and albumin [ALB] measured on the (**I**) Vetscan and (**J**) Piccolo rotors. Red/pink circles show 0.9% saline-treated NHPs, and blue/light blue triangles show rVSV-MARV-vaccinated NHPs. Solid red or blue bars denote Experiment 1, black bars Experiment 2, where the height is equal to the geometric mean of the group. Error bars denote geometric standard deviation. Day 10 pc Piccolo values are only available for Experiment 2 (unavailable cohort data shown as #). A two-tailed unpaired *t*-test compared 0.9% saline and rVSV-MARV groups on a specific day (* *p* < 0.05, ** *p* < 0.01, *** *p* < 0.001).

**Table 1 viruses-16-01181-t001:** C Reactive Protein Levels.

Day Post Challenge ^2^	Experiment 1: 0.9% Saline ^1^	Experiment 1: rVSV-MARV	Experiment 2: 0.9% Saline	Experiment 2: rVSV-MARV2
0	<5, <5, <5	<5, <5, <5, <5, 8.2	<5, <5, <5	<5, <5, <5, <5, <5
7	~~, 27.5, ~~	<5, <5, 126, <5, <5	26.6, 106, 33.2	5.3, <5, 6.3, <5, <5
10	N/A	ND	N/A	<5, <5, <5, <5
28	N/A	ND	N/A	5.1, <5, <5, <5

N/A—samples not available, ND—samples not taken for analysis. ^1^ Individual values for each sample within the cohort, in mg/L. LLOQ < 5 mg/L. “~~” denotes a reading outside the assay linear range of 5–200 mg/L. ^2^ Data for days 10 and 28 post challenge carried out for Experiment 2 only, assay not run for UG631 on day 28.

## Data Availability

Data may be made available upon request.

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
