# Peer review of "Efficacy and Immunogenicity of a Recombinant Vesicular Stomatitis Virus-Vectored Marburg Vaccine in Cynomolgus Macaques"

_viruses, 2024, doi:10.3390/v16081181_

Round 1

Reviewer 1 Report

Comments and Suggestions for Authors

In this paper, Vidyleison Neves Camargos and colleagues, are demonstrating the immunogenicity and protective efficacy of a vaccine against a deadly infection by Marburg virus (MARV). The authors are using a recombinant Vesicular Stomatitis Virus where the native glycoprotein is replaced by the glycoprotein of MARV (rVSVdG-MARV). This vaccine was previously used in guinea pig model demonstrating protective capacity. MARV is still a worldwide concern and of great interest for pandemic preparedness programs. Here, the authors are describing how two different doses of rVSVdG-MARV vaccine protect non-human primates from deadly infection when administrated as a single shot 42 days prior receiving MARV.

General comments

1) The authors are testing two different doses of their vaccine rVSVdG-MARV at 6.7x107pfu and 1.6x107 pfu. Was there an expectation that those two doses will provide different potency of  immune responses? How those two doses were selected, as most of the publications on vaccination with rVSV-MARV are using 1x107 pfu. I would suggest the authors to add a comment on their dose selection as they could have selected a lower dose and still providing 100% protection.

2) As no major difference in antibodies nor neutralizing titers were observed across experiments 1 and 2, and taking into account that the vaccination was performed with two different lots at two different doses, I am wondering if the two lots of vaccine where similar (for example, Experiment 2 lot being more potent than Experiment 1 lot). Did the authors performed some preliminary testing of the vaccine in-vitro to demonstrate that those two lots were equivalent to express the antigen?

Specific comments

1) Line 54, page 2: “replicates to high titers in vitro” are the authors referring to production capability? or using that vector as tool in an in-vitro assay? If not, I will suggest the authors to rephrase or temper that statement, as for vaccine, high replication viral vector is not preferred for safety concerns and generally an attenuated form of the vector is used with limited replication.

2) Figure 3, page 8: one comment will be to break the x-axis between non-irradiated and gamma irradiated caption or to stop the LLOQ cutoff dashed line for the gamma irradiated portion of the graph. As the authors have very well controlled their experiments, they have observed that the gamma irradiation impacted their samples and in consequence, I am not sure that the defined LLOQ using non irradiated samples applies to gamma irradiated samples as it seems that the background of the assay increased.

3) Figure 7, page 11: In experiment 2, for the decrease in virus titer observed at day 7 post-challenge, compared to day 5 and day 8 which where both higher, did the authors have re-extracted and remeasured their samples to confirm that unexpected decrease? This is a rather unusual pattern.

4) Line 426, page 12: The authors are mentioning “saline-treated mice” which must be an error as only non-human primates were used in that study.

5) Line 496, page 13: Please correct the typo “rVS-MARV”.

6) Lines 633 to 640, page 18: Even if the point addressed in this paragraph is of interest and already covered by other authors, I am not sure of its relevance for this particular paper. The authors are mentioning that the rVSV vectors are “significantly attenuated and does not replicate in these transmission vectors (aka flies and midges)”. So if rVSV cannot transmit via those insects why raising the concern in the first place? I might have missed the point that the authors were trying to make so I will suggest the authors to either rephrase or remove this section.

Author Response

General comments

  1. The authors are testing two different doses of their vaccine rVSVdG-MARV at 6.7x107pfu and 1.6x107 pfu. Was there an expectation that those two doses will provide different potency of immune responses? How those two doses were selected, as most of the publications on vaccination with rVSV-MARV are using 1x107 pfu. I would suggest the authors to add a comment on their dose selection as they could have selected a lower dose and still providing 100% protection.

Response: We thank the reviewer for this important question. These doses were chosen due to dose-dependent efficacy of rVSV-MARV and many other vaccines. As Experiment 1 was the first proof of concept NHP protection study, the highest vaccine dose was used to ensure efficacy. Also, scaling up the dosing when moving from smaller animal models to NHPs is a strategy that had also been used by other groups too. Our vaccine candidate was previously tested in Guinea pigs immunized with a high dose of 2 x 106 PFU (https://www.mdpi.com/2076-393X/10/7/1004) and it provided 100% protection against a lethal MARV challenge, but when using a lower dose of 2 × 104 PFU, protection was also high but not 100% effective as observed with the high dose. Jones et al (2005) used a similar strategy by using a inoculation dose three logs greater than the doses successfully used to immunize mice and guinea pigs against ZEBOV when testing their rVSV-based vaccine candidate against EBOV/MARV (https://www.nature.com/articles/nm1258). Once all data was collected and protection against MARV challenge was verified, we started the second study using a dose level consistent with our clinical experience with rVSV-vectored vaccines and the literature. Future vaccine down-dosing studies in NHPs are planned.

  1. As no major difference in antibodies nor neutralizing titers were observed across experiments 1 and 2, and taking into account that the vaccination was performed with two different lots at two different doses, I am wondering if the two lots of vaccine where similar (for example, Experiment 2 lot being more potent than Experiment 1 lot). Did the authors performed some preliminary testing of the vaccine in-vitro to demonstrate that those two lots were equivalent to express the antigen?

Response: We understand the reviewer’s concern about this deviation. We would like to point out that both vaccine lots were made from the same working viral seeds. Lot#PTR4.1-15Oct2020 was generated as a late development lot and Lot#155941 was an engineering lot manufactured just prior to GMP production. Both lots were potent for infectious titer as measured by the plaque assay and consistent with product specifications. In addition, the product is genetically stable as determined by postproduction next generation sequencing (NGS). Lot-to-lot variation is predominantly related to virus titer and not the virus phenotype.

Specific Comments:

1) Line 54, page 2: “replicates to high titers in vitro” are the authors referring to production capability? or using that vector as tool in an in-vitro assay? If not, I will suggest the authors to rephrase or temper that statement, as for vaccine, high replication viral vector is not preferred for safety concerns and generally an attenuated form of the vector is used with limited replication.

Response: We thank the reviewer for this suggestion. In the manuscript, we wanted to highlight the replication to high titers in vitro as a benefit for manufacturing rVSV-based vaccines compared to other viral vectored platforms. Thus, we modified the sentence in line 54-55, page 2 (added text in bold below): “Of all approaches, the VSV-vectored platform is one of the most attractive as it combines an impressive safety profile, strong antigen presentation, replicates to high titers in vitro, which is manufacturing advantageous compared to other viral-vectored vaccines, induces strong innate and adaptive immunity, and has been effective as a strategy for a variety of viral diseases (reviewed in [7])” 

2) Figure 3, page 8: one comment will be to break the x-axis between non-irradiated and gamma irradiated caption or to stop the LLOQ cutoff dashed line for the gamma irradiated portion of the graph. As the authors have very well controlled their experiments, they have observed that the gamma irradiation impacted their samples and in consequence, I am not sure that the defined LLOQ using non irradiated samples applies to gamma irradiated samples as it seems that the background of the assay increased.

Response: We agree and thank the reviewer for this suggestion. We have edited the Figure 3 graph and stopped the LLOQ dashed line indicating this limit only for non-irradiated samples. We also updated the figure legend to reduce any ambiguity.

3) Figure 7, page 11: In experiment 2, for the decrease in virus titer observed at day 7 post-challenge, compared to day 5 and day 8 which where both higher, did the authors have re-extracted and remeasured their samples to confirm that unexpected decrease? This is a rather unusual pattern.

Response: We thank the reviewer for pointing out this potential deviation. Unfortunately, we do not have stored sera from this specific time point (7 days post challenge) to repeat the RNA extraction and RT-qPCR assays to directly address the reviewer’s question. However, we would like to highlight that MARV viral RNA from all serum samples in Experiment 2 were quantified by RT-qPCR assay in two different plates that had been included the same standard positive controls in both plates at various dilutions, which resulted in very similar Ct values between plates. Additionally, since (i) these samples were all processed at the same time and in similar laboratory conditions as best as possible, (ii) that there was no statistical difference between the viral RNA levels between days 5, 7, and 8 post challenge in the Experiment 2, we had no indication that a technical error occurred. So, we assumed that this difference was a normal variation in the experimental model rather than an issue with the quantification methods used in our work. Viremia levels are often variable and the main point in this analysis was to show that the controls were viremic and the vaccinated NHPs were not.

4) Line 426, page 12: The authors are mentioning “saline-treated mice” which must be an error as only non-human primates were used in that study.

Response: We thank the reviewer for this observation. This was indeed an error and the word “mice” was corrected to “NHPs”.

5) Line 496, page 13: Please correct the typo “rVS-MARV”.

Response: We thank the reviewer for this observation. We corrected the typo by changing the word to “rVSV-MARV”

6) Lines 633 to 640, page 18: Even if the point addressed in this paragraph is of interest and already covered by other authors, I am not sure of its relevance for this particular paper. The authors are mentioning that the rVSV vectors are “significantly attenuated and does not replicate in these transmission vectors (aka flies and midges)”. So if rVSV cannot transmit via those insects why raising the concern in the first place? I might have missed the point that the authors were trying to make so I will suggest the authors to either rephrase or remove this section.

Response: We thank the reviewer for this suggestion. However, we included this topic in the discussion because wild-type VSV transmission is related to two main factors: host susceptibility and the virus titer. During non-clinical and clinical development vaccinemia and shedding is always monitored due to safety concerns. Since the possibility of shedding into the environment or transmission via arthropods is always monitored, low vaccinemia provides some confidence this vaccine would not be transmitted. Therefore, we believe that this section could be of interest to the readers and should remain included.

Reviewer 2 Report

Comments and Suggestions for Authors

Review of Manuscript “Efficacy and Immunogenicity of a Recombinant Vesicular Stomatitis Virus-Vectored Marburg Vaccine in Cynomolgus Macaques “ by Vidyleison Neves Camargos et al.. 

In the present manuscript the authors report on experiments examining the protective effects of a rVSV-based vaccine expressing the Marburg Virus glycoprotein (rVSV-MARV), which was initially tested in guinea pigs, in the well-characterized cynomolgusmacaque MARV infection model. The authors could demonstrate that a single dose of the rVSV-MARV vaccine with as little as 1.6E7 pfu protected all animals included in the study against the lethal MARV infection. In line with these observations, antibody responses including neutralizing ones were established within 14 days. During this time period, the vaccinemia noted in the first days post vaccination could also be resolved, which is an important point regarding a possible transmission of the vaccine by blood-feeding insects. In addition to clinical signs for MARV disease progression, a broad range of hematological and clinical parameters were examined and showed no alterations in the vaccinated animals.

The manuscript is well-written and the data obtained from the extensive investigations is presented very clearly. My only point of criticism is that it did not become quite clear to me, what may be the advantages or also disadvantages of the present rVSV-MARV vaccine as compared to those described in earlier publications quoted by the authors.

Author Response

Thank you for the rapid and thorough review of our manuscript. Below are our responses to the reviewer’s inquiries.

General Comment: The manuscript is well-written and the data obtained from the extensive investigations is presented very clearly. My only point of criticism is that it did not become quite clear to me, what may be the advantages or also disadvantages of the present rVSV-MARV vaccine as compared to those described in earlier publications quoted by the authors.

Response: We thank the reviewer for the comments. The differences between this rVSV-MARV vaccine candidate and others mentioned throughout the manuscript are related to the development phase and controls of vaccine production and release. Our vaccine candidate was manufactured and tested by highly characterized and qualified procedures typically not employed in a research laboratory. Furthermore, this vaccine candidate was developed and manufactured in compliance with FDA requirements for human clinical testing. Non-PHV rVSV Marburg vaccine candidates described in the literature and referenced in this manuscript were made in research laboratories.

An example of PHV’s vaccine differentiation is that the PHV vaccine candidate was cloned by 5 consecutive rounds of plaque purification. After amplification a single clone was selected for Master Virus Seed (MVS) production having met the following criteria:

  • NGS showed no amino acid mutations in the MARV GP gene and ≤10% single nucleotide polymorphisms frequency compared to the uncloned Research Virus Seed
  • Acceptable growth and productivity in Vero cells
  • No change in in vitro phenotype (growth kinetics/CPE, plaque size and morphology)
  • Acceptable test results- Potency (titer), identity (NGS), sterility, Mycoplasma (PCR).

The cloned virus was chosen and used to prepare the MVS and subsequently working viral seed (WVS) under GMP conditions.  These seeds are essential starting materials used to produce development, engineering and subsequently GMP vaccine lots.  A later development lot and engineering lot were used in this manuscript.

Again, thank you for your time and complete review this manuscript. Your suggestions have improved the quality of this work.

Reviewer 3 Report

Comments and Suggestions for Authors

The manuscript by Camargos et al reports the evaluation of the efficacy and immunogenicity of a vaccine based on the rVSV expressing the Marburg virus GP in Cyanomolgus macaques.

The manuscript is well written and detailed in material and methods description as well as in the reported results.

Data are important in the field of vaccines based on VSV against viral hemorrhagic fever viruses and the article deserves to be published.

I have only minor comments:

Line 29, please check the sentence to be sure that it is correct

Add a comment regarding the choice of the two doses used for the MARV challenge.

Comments on the Quality of English Language

Quality is high, there are very few/minor editing modifications.

Author Response

Thank you for the rapid and thorough review of our manuscript. Below are our responses to the reviewer’s inquiries.

General Comment

  1. Line 29, please check the sentence to be sure that it is correct

Response: We thank the reviewer for observing this error. We changed the sentence from “(from which tissues were not collected and no causal link was not established)” to “(from which tissues were not collected and no causal link was established)”.

  1. Add a comment regarding the choice of the two doses used for the MARV challenge.

Response: We thank the reviewer for commenting on the dosing choice. As study 1 was the first proof of concept NHP protection study, the highest vaccine dose was used to ensure efficacy. The second study used a dose level consistent with our clinical experience with rVSV-vectored vaccines and the literature. Future vaccine down-dosing studies in NHPs are planned.

Again, thank you for your time and complete review this manuscript. Your suggestions have improved the quality of this work.
